# Transcription induces context-dependent remodeling of chromatin architecture during differentiation

**Sanjay Chahar[1]⊙, Yousra Ben Zouari[1]⊙, Hossein Salari[2], Dominique Kobi[1], Manon Maroquenne[1], Cathie Erb[1], Anne M. Molitor[1], Audrey Mossler[1], Nezih Karasu[1], Daniel Jost[2], Tom Sexton[1]***

**1** University of Strasbourg, CNRS, Inserm, IGBMC UMR 7104-UMR-S 1258, Illkirch, France, **2** Laboratoire de Biologie et Modélisation de la Cellule, École Normale Supérieure de Lyon, CNRS, UMR5239, Inserm U1293, Université Claude Bernard Lyon 1, Lyon, France

⊙ These authors contributed equally to this work.
* sexton@igbmc.fr

**Data Availability Statement:** All sequencing data generated in this study are available at GEO with the accession GSE218090. New code used in this manuscript is provided in Github: https://github.

## Abstract

Metazoan chromosomes are organized into discrete spatial domains (TADs), believed to contribute to the regulation of transcriptional programs. Despite extensive correlation between domain organization and gene activity, a direct mechanistic link is unclear, with perturbation studies often showing little effect. To follow chromatin architecture changes during development, we used Capture Hi-C to interrogate the domains around key differentially expressed genes during mouse thymocyte maturation, uncovering specific remodeling events. Notably, one TAD boundary was broadened to accommodate RNA polymerase elongation past the border, and subdomains were formed around some activated genes without changes in CTCF binding. The ectopic induction of some genes was sufficient to recapitulate domain formation in embryonic stem cells, providing strong evidence that transcription can directly remodel chromatin structure. These results suggest that transcriptional processes drive complex chromosome folding patterns that can be important in certain genomic contexts.

## Introduction

Eukaryotic genomes are spatially highly organized, permitting access of transcriptional machinery to the appropriate loci despite extensive compaction for containment within the nucleus [1]. A key architectural feature of metazoan chromosomes is their organization into autonomously folded domains, termed "topologically associated domains" (TADs), whose arrangement correlates very well with functional demarcation of chromatin regions according to gene expression, epigenetic marks, and replication timing [2–4]. TADs have been proposed to ensure appropriate gene expression by limiting the functional range of distal transcriptional enhancers, evidenced by pathologies due to ectopic gene activation on loss of TAD borders [5]. TADs may conversely facilitate enhancer activity on their cognate genes by limiting their

com/TomSexton00/Chahar_etal_analysis and is available on Zenodo: doi 10.5281/zenodo. 10060152. Source data are available in S1 Data.

**Funding:** Work in the TS lab was supported by funds from the European Research Council (ERC) under the European Union's Horizon 2020 research and innovation program (Starting Grant 678624 - CHROMTOPOLOGY), the ATIP-Avenir program, and the grant ANR-10-LABX-0030-INRT, a French State fund managed by the Agence Nationale de la Recherche under the frame program Investissements d'Avenir ANR-10-IDEX-0002-02. YBZ was additionally supported by la Region Grand Est. Work in the DJ lab was supported by funding from Agence Nationale de la Recherche (ANR-18-CE12-0006-03; ANR-18-CE45-0022-01). The funders had no role in study design, data collection and analysis, decision to publish, or preparation of the manuscript.

**Competing interests:** The authors have declared that no competing interests exist.

**Abbreviations:** CHESS, Comparison of Hi-C Experiments using Structural Similarity; CRISPRa, CRISPR activation; DN3, double negative; DP, double positive; ESC, embryonic stem cell; TAD, topologically associated domain; VPR, VP64-p65-Rta.

effective search space within a domain [6]. However, experimental perturbations causing acute, extensive loss of TADs have only modest effects on the transcriptome [7–9], suggesting that chromatin topology is not necessarily instructive in gene regulation. Alternatively, by stabilizing particular configurations more or less favorable for transcriptional firing, TADs could serve to reduce transcriptional noise and cell-to-cell variability [10]. Metazoan TAD borders are highly enriched for binding of the insulator protein CCCTC-binding factor (CTCF) [2], with the curious feature that flanking TAD borders predominantly comprise CTCF motifs in convergent orientation [11]. A popular model for TAD formation is loop extrusion, whereby the ring-like cohesin complex binds and translocates bidirectionally along chromatin, bringing linearly distal regions near to one another. The "collision" of convergent CTCF-bound sites with cohesins is proposed to stall loop extrusion, thus creating metastable interactions between TAD boundaries [12–15]. Live imaging experiments have visualized such predicted interactions, although they are relatively infrequent and transient [16]. Inversion of CTCF motifs disrupts chromatin interactions [12,17,18], highlighting the importance of CTCF orientation in chromatin architecture. However, the prevailing absence of new interactions between CTCF pairs now brought into convergent orientation, as well as the existence of CTCF-independent spatial chromatin domains [7,19], suggests that other factors participate in their formation [20].

Another feature predominantly enriched at TAD boundaries is active genes [2,3,21]. A priori, RNA polymerase promoter binding and subsequent tracking along chromatin during transcription may cause topological disruption of the underlying DNA fiber [22]. It has thus been proposed that RNA polymerase binding and/or transcription may organize TADs, either directly or by acting as a CTCF-independent roadblock to loop extrusion [23,24]. However, conflicting reports mean it is currently unclear to what extent transcription modulates chromatin architecture and whether any such effects are direct. For example, the inactive X chromosome lacks TADs globally except around the few genes escaping silencing [25], yet transcriptionally inert sperm chromosomes maintain essentially all TADs [26,27]. One possible explanation for the discrepancy is that the noncoding RNA *Xist* excises cohesin from the inactive X chromosome [28], so loop extrusion processes could be perturbed chromosome-wide independently of transcriptional effects. TADs arise in early embryogenesis of mammals and flies, coinciding with the onset of zygotic gene activation, but the process is largely unaffected on treatment with drugs inhibiting transcription [26,29,30]. A different study in a cell line reported that TADs were weakened on similar pharmacological transcription inhibition [31]. Looking more mechanistically, a controlled study of cohesin-mediated loop extrusion showed that this ATP-dependent process can occur independently of ongoing transcription [32], but this does not preclude other means of architectural modulation. For example, transcription, among other mechanisms, appears to influence cohesin loading onto chromatin [33], so may indirectly influence loop extrusion processes in some contexts. Alternatively, viral-induced extension of transcription beyond usual termination sites was found to disrupt spatial chromatin domains, perhaps due to removal of CTCF by the engaged RNA polymerase and, hence, loss of boundary function [34]. As ultrahigh-resolution chromatin interaction (Micro-C) maps of mammalian chromosomes became available, small spatial chromatin domains at the level of single expressed genes began to be discerned [35], reminiscent of what was previously observed in yeast [36] and transcription-linked "compartmental domains" described in *Drosophila* [37]. This suggests two, nonmutually exclusive means by which gene activity can modulate chromatin architecture. Firstly, chromatin tends to compartmentalize into coassociated active "A" compartments, separate from coassociated silent "B" compartments [38], a process that is independent or perhaps even antagonistic to loop extrusion-mediated TAD formation [7–9]. Thus, small active genes within large regions of inactive chromatin could indirectly form a domain boundary by disrupting its resident B compartment [37].

Secondly, RNA polymerase tracking could directly compact the underlying transcription unit to generate its own spatial chromatin domain, reminiscent of previous reports that the 5′ and 3′ ends of active genes interact to assure transcription directionality [39]. In support of both of these phenomena, analysis of Micro-C data in mouse embryonic stem cells (ESCs) revealed that intragene contact frequency correlates with RNA polymerase occupancy of the gene and that RNA polymerase binding, while being a poor predictor of domain boundary location, is a reasonably good predictor of boundary strength [35]. Despite these promising findings, the most direct tests of transcription perturbation (not just those relying on inhibitor drugs, which may have secondary effects) have shown rather limited effects on chromatin structure. Acute ablation of any of the 3 RNA polymerases with an auxin-inducible degron had negligible effects on any feature of genome architecture, including TADs [40]. The resolution of this study may not have been high enough to discern subtle effects, and a similar experimental approach showed that whereas interphase chromatin was indeed largely unaltered, resetting of TADs just after mitosis was affected by loss of RNA polymerase II [41]. As an alternative means of testing the direct effect of transcription on chromatin topology, another study used CRISPR activation (CRISPRa) [42] to ectopically induce 2 specific genes in ESCs: *Sox4* and *Zfp608* [21]. These genes were found to become new domain boundaries on differentiation to neural precursors, concomitant with their transcriptional activation, although their induction in ESCs was unable to cause any topological changes at the target loci. Collectively, this body of work suggests that, despite extensive correlation between spatial domain organization and gene activity, it remains unclear whether there is a direct causative role for transcription in chromatin topology, and any such links are likely to be subtle and limited to specific genomic contexts.

In this study, we assessed the developmental dynamics of spatial chromatin domain organization, using mouse thymocyte development as a model system. Although domain structures were largely conserved, in line with previous lower-resolution studies, we observed some specific remodeling events that coincided with transcriptional changes. Notably, we observed apparent broadening of a TAD border at the *Bcl6* gene, concomitant with extension of pause-released RNA polymerase into the gene body. Moreover, we observed cases of spatial chromatin domains corresponding to single activated genes, lending further support to the previously mentioned models by which transcription could modulate chromatin topology. Most importantly, for the tested genes, *Nfatc3* and *Il17rb*, CRISPRa induction of the gene in ESCs was sufficient to recapitulate the thymocyte architectural remodeling event, providing, to our knowledge, the first direct evidence that transcription can be a driver of chromosome folding in certain contexts.

## Results

### High-resolution chromatin architecture at key thymocyte genes

To assess at higher resolution whether spatial chromatin domains may be selectively remodeled around genes that are differentially expressed on developmental transitions, we performed Capture Hi-C [43,44] on mouse CD4⁻ CD8⁻ CD44⁻ CD25⁺ (double negative; DN3) and CD4⁺ CD8⁺ (double positive; DP) thymocytes, representing cells just initiating and cells just after the beta-checkpoint for productive T cell receptor beta-chain rearrangement [45]. We used tiled capture oligonucleotides spanning *Dpn*II fragments across 8 approximately 600 kb regions, centered on key thymocyte genes located close to TAD borders, identified from Hi-C maps generated in mouse ESCs [21]: three that are up-regulated on DN3-to-DP transition (*Bcl6*, *Nfatc3*, *Rag1*), three that are down-regulated (*Cdh1*, *Il17rb*, *Pla2g4a*), and two whose expression are relatively unchanged (*Cd3*, *Zap70*) (**S1A Fig**; **S1** and **S2 Tables**). We also performed

conventional Hi-C on DN3 and DP cells for genome-wide but lower-resolution assessment of any chromatin topology changes (overview of datasets in this study given in **S3 Table**). Capture Hi-C was additionally performed on ESCs to provide a reference point and comparison with high-resolution conventional Hi-C data [21]. As expected, the Capture Hi-C provided improved resolution at the target regions compared to previously generated Hi-C maps for thymocytes [46], even though the latter dataset had approximately 4-fold greater sequencing depth (**Fig 1A**). Biological replicates were highly correlated (Spearman correlation coefficient ≥0.91; **S1B and S1C Fig**; **S4 Table**) and were pooled for improved resolution in subsequent analyses. The resultant maps were of sufficient quality to resolve chromatin interactions between promoters and putative enhancers, marked with acetylation of lysine-27 on histone H3 (H3K27ac), and between CTCF-bound sites, further confirmed by 4C-seq analysis (**Figs 1B–1D** and **S1D**). Visual inspection of the Capture Hi-C maps showed that thymocytes seem to have better defined TADs and more heterogeneous structures than ESCs (**Figs 1B** and **S2–S7**), in line with observations that ESC chromatin is generally more open and "plastic" than differentiated cells [47]. Observing the contact strength decay of the Capture Hi-C datasets with increasing genomic distance reinforces these trends by showing greater variability of contact strength at a given genomic separation within thymocytes (**Fig 1E**). Further, the *cis*-decay profiles for both Capture Hi-C and, at lower resolution, the genome-wide Hi-C, show that DP chromatin is globally more compact than the other cell types, in agreement with previous observations [48]. Overall, these results validate the quality of the Capture Hi-C datasets for subsequent in-depth comparisons across cell types.

## Chromatin topology is largely conserved but with cell type–specific differences

Visual inspection of Capture Hi-C maps around the target regions suggested that spatial domain organization was largely conserved across the studied cell types, and particularly in between the 2 thymocyte populations (**S2–S7 Figs**), in line with lower-resolution studies [46,49]. To compare TAD organization more systematically, we computed insulation scores [50] at 5 kb resolution within the regions targeted by capture oligonucleotides. Compared to other methods that just call the positions of domains [51], insulation has the advantage of giving a quantitative border "score" for all genomic intervals. Since chromosomal domains are known to be nested and hierarchical [52,53], we computed insulation over multiple sliding window widths to alter sensitivity to domains of different size. In agreement with previous observations that TADs appear to be better defined in differentiated cells than in pluripotent cells, insulation scores are more homogeneous in ESCs, whereas more striking insulation score minima (representing strongly insulating borders) and maxima (representing the centers of well-defined folded domains) are apparent in thymocytes (**Fig 2A**). Over a wide range of window sizes used to compute insulation score, the more homogeneous distribution in ESCs is significantly different to those of the thymocytes (two-sided Kolgorov–Smirnov test; **S5 Table**). However, the genomic location of TAD border candidates is well conserved across cell types, since insulation scores are overall very well correlated (Spearman correlation coefficient between 0.71 and 0.91; **Fig 2B** and **S5 Table**). These observations are supported by analysis of the lower-resolution Hi-C data, both at the specified capture regions and genome-wide (**S8A and S8B Fig** and **S6 Table**). Thus, on first analysis, domain border *location* appears to be well conserved across cell types, but overall, domain *strength* is higher in differentiated cells than in pluripotent ESCs.

To explore further whether there are any cell type–specific differences in TAD organization, we performed principal component analysis on the insulation scores. Across all tested

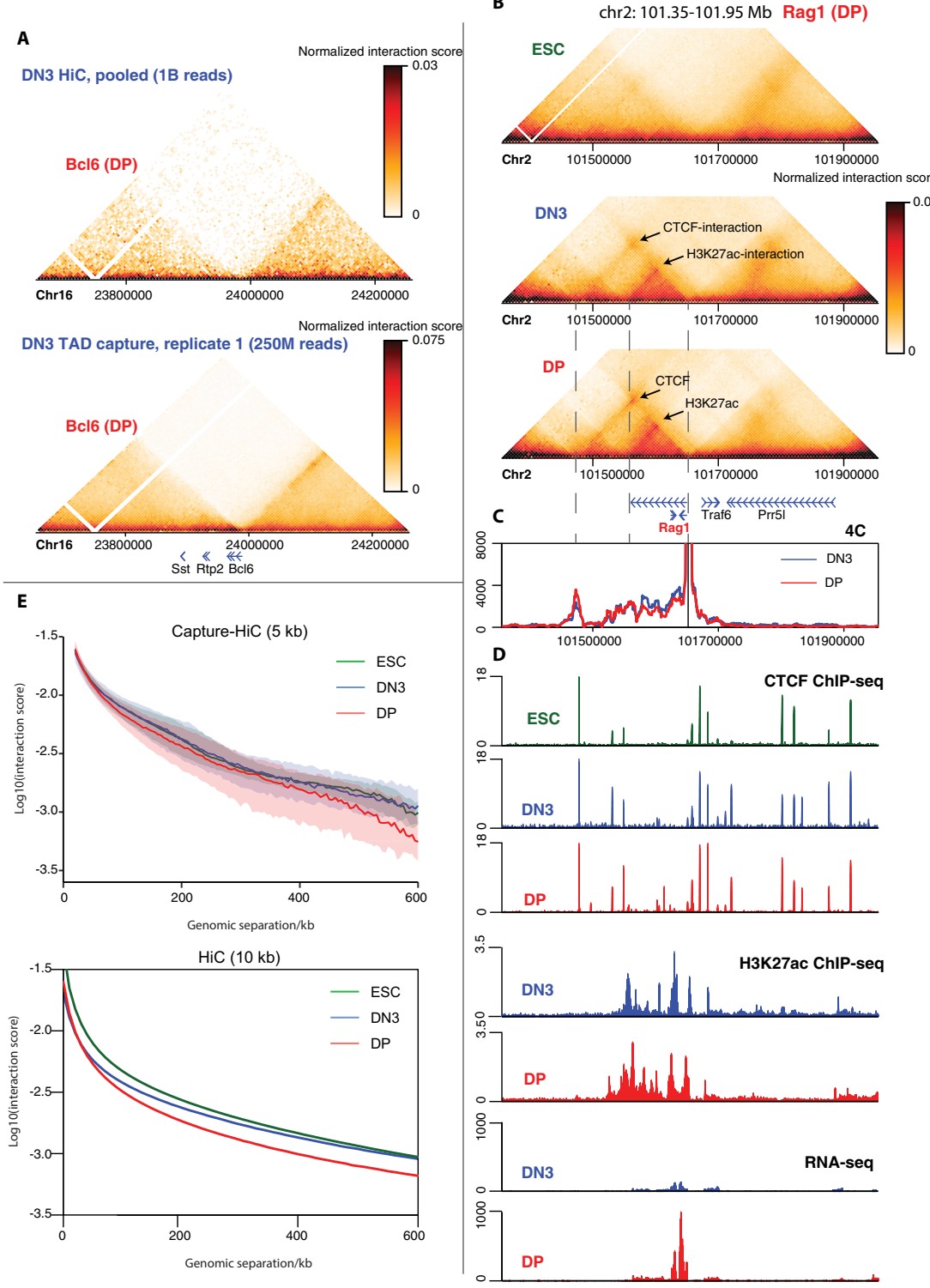

**Fig 1. High-resolution interrogation of chromatin architecture in thymocytes with Capture Hi-C.** (**A**) Maps of DN3 chromatin interactions for an approximately 600 kb region around the *Bcl6* gene from pooled Hi-C [46] (approximately 1 billion reads; top) or one replicate of Capture Hi-C (approximately 250 million reads; bottom) are shown at 10 kb and 5 kb resolution, respectively. Positions of genes are denoted underneath the maps. (**B**) Pooled Capture Hi-C interaction maps for an approximately 600 kb region around the *Rag1* gene are shown for ESCs, DN3, and DP cells at 5 kb resolution. Positions of genes

are denoted underneath the maps. Arrows denote thymocyte-specific interactions. (**C**) 4C-seq profiles, using the *Rag1* promoter as bait, performed in DN3 (blue) and DP (red) cells. Dashed lines denote position of *Rag1* promoter and selected CTCF or H3K27ac peaks, which form the bases of the interactions highlighted in (**B**). (**D**) Epigenomic profiles at the same genomic location as shown in (**B**) and (**C**). Top to bottom: ChIP-seq tracks (normalized as counts per million reads) for CTCF in ESCs (green), DN3 (blue), and DP (red) cells; ChIP-seq tracks (normalized as counts per million reads) for H3K27ac in DN3 and DP cells; RNA-seq (non-strand-specific; normalized as counts per million reads) in DN3 and DP cells. (**E**) Plot of median normalized interaction score against genomic separation from the pooled Capture Hi-C (top) and Hi-C (bottom) datasets, at 5 kb and 10 kb resolution, respectively, for ESCs (green), DN3 (blue), and DP (red) cells. For the Capture Hi-C data, shading denotes the interquartile range to indicate the variability of these *cis*-decay distributions. Source data can be found in S1 Data. DN3, double negative; DP, double positive; ESC, embryonic stem cell.

insulation score window sizes, biological replicates clustered closely together, and the 3 cell types were clearly distinct from each other, suggesting that cell type–specific architectural patterns are indeed present (**Figs 2C** and **S8C**). Such differences quantified by the principal component analysis could be due to gain/loss of specific borders to generate new domains, altered strengths of preexisting TADs (which has been observed for some genes at early thymocyte transitions [46]), and/or accumulated small but reproducible changes dispersed across the studied genomic regions, which individually make no major difference to the Hi-C maps. Since insulation scores from biological replicates clustered so closely together, we pooled the Capture Hi-C replicates and computed higher-resolution insulation scores over 2 kb genomic bins, finding strong local minima maintained across different sliding window sizes to call the most robust borders. We identified approximately 50 borders within the captured regions for each cell type (92 total; **S7 Table**). In line with the above observations, many of the strongest borders were conserved throughout development, but cell type–specific/strengthened domains were apparent, in particular when comparing thymocytes to ESCs (**Figs 2D–2G** and **S8D**). As may be expected from their developmental proximity, the 2 thymocyte populations had more borders in common (DN3-DP Jaccard index 0.57) than with ESCs (DN3-ESC Jaccard index 0.43; DP-ESC Jaccard index 0.32). Applying an analogous border calling method to the Hi-C data at lower (10 kb) resolution gave similar results genome-wide (**S9 Fig** and **S8 Table**), although the separation between thymocytes and ESCs was not as pronounced. This may be due to the focus of the Capture Hi-C around thymocyte-specific genes, which might be expected to have greater architectural differences than more general, housekeeping genes. Overall, we find a general conservation of spatial domain organization between different cell types, although higher-resolution analysis can uncover cell type–specific insulation patterns, sometimes associated with variant borders.

## Pause-released polymerase may broaden spatial domain boundaries

We next further investigated the cell type–specific chromatin architectures uncovered by the Capture Hi-C approach. When comparing ESCs to thymocytes, we found cases of thymocyte-specific subdomains arising from establishment or strengthening of boundaries near the promoter of genes expressed specifically in thymocytes, both in Capture Hi-C (e.g., the *Cd3* cluster; **Figs 2D–2F** and **S5**) and lower-resolution Hi-C (e.g., *Runx1*; **S9C Fig**) maps. In these described examples, the new boundaries were accompanied by increased binding of CTCF near the promoter during thymocyte development. The downstream boundary appeared to be maintained across the studied cell types, corresponding with a conserved CTCF binding site. Based on the loop extrusion model, these developmental architectural changes between ESCs and thymocytes could be explained solely by differential CTCF binding. Whether underlying gene activation could be a cause or consequence of 3D genome structure, if functionally linked at all, thus remains unclear from these cases. However, interrogating differences between DN3 and DP chromatin structures, which have very similar CTCF binding profiles, uncovered

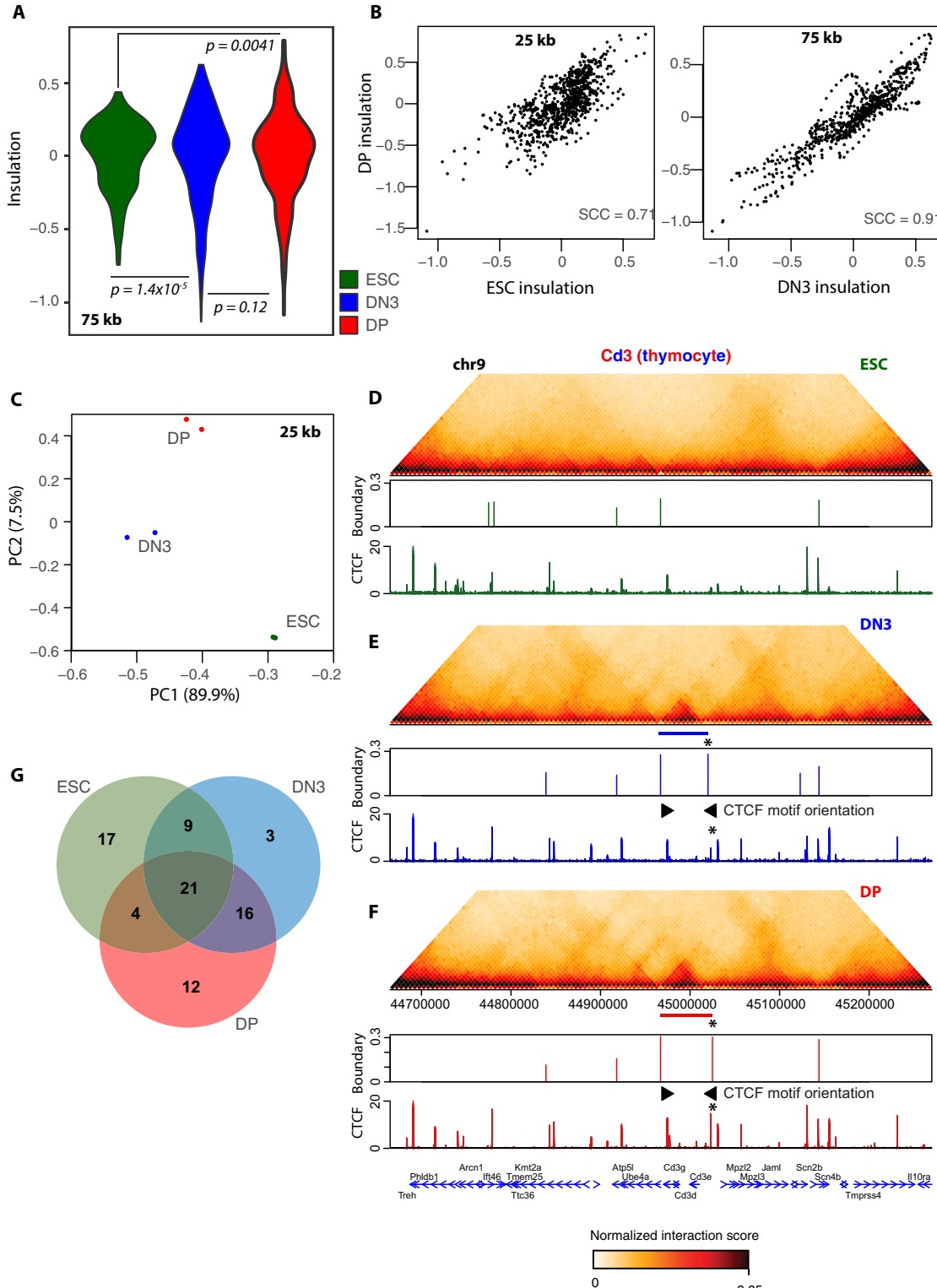

**Fig 2. TADs are largely conserved, with cell type–specific differences. (A)** Violin plots for distributions of insulation scores computed on pooled Capture Hi-C datasets at 5 kb resolution, using a 75-kb (15 bin) window, on ESCs (green), DN3 (blue), and DP (red) cells. The more homogeneous ESC distribution is significantly different to DN3 ($p = 1.4 \times 10^{-5}$) and DP ($p = 0.0041$) cells, but the 2 thymocyte distributions are not significantly different to each other ($p = 0.12$; Kolmogorov–Smirnov test). **(B)** Scatter plots of insulation scores computed on pooled Capture Hi-C datasets at 5 kb resolution, using a 25-kb (5 bin; left) or 75-kb (15 bin;

right) window, comparing DP cells with ESCs (left) or DN3 cells (right). Spearman correlation coefficients (SCC) are denoted. (**C**) Plot of first 2 principal components for insulation scores computed on biological replicates of Capture Hi-C datasets at 5 kb resolution, using a 25-kb (5 bin) window for ESCs (green), DN3 (blue), and DP (red) cells. Amount of variance accounted for in each principal component is denoted in the axis labels. (**D-F**) Pooled Capture Hi-C interaction maps for an approximately 600 kb region around the *Cd3e* gene are shown for (**D**) ESCs, (**E**) DN3, and (**F**) DP cells, above plots showing the positions and scores of called TAD boundaries and ChIP-seq profiles (normalized as counts per million reads) for CTCF in the same cells. Blue and red bars denote the position of a thymocyte-specific TAD, which is accompanied by the gain of boundary at the 5′ end, which is bound by more CTCF (denoted by asterisk). Motif orientations of CTCF sites bordering the new TAD is shown with arrows. Positions of genes are shown below the plots. (**G**) Venn diagram showing overlapping of called TAD boundaries for the pooled Capture Hi-C datasets of ESCs (green), DN3 (blue), and DP (red) cells. Source data can be found in S1 Data. DN3, double negative; DP, double positive; ESC, embryonic stem cell; TAD, topologically associated domain.

potentially more direct links between transcription and chromatin architecture. Firstly, a strong, conserved border at the *Bcl6* promoter is altered in DP cells where the gene is highly expressed: An additional weak "stripe" extending the upstream spatial domain indicates the presence of a secondary broader border, adjacent to the major sharp boundary at the promoter (see rectangle and arrow in **Figs 3A** and **S7**). Notably, the domain "extension" corresponds with the *Bcl6* coding region, and ChIP-seq data show that RNA polymerase II is paused at the promoter/major boundary in DN3 cells while being released to fully transcribe *Bcl6* in DP cells (**Fig 3B** and **3C**). These Capture Hi-C data are thus consistent with a model whereby bound and/or engaging RNA polymerase can act as a topological boundary itself, perhaps by stalling loop extrusion processes, as has been reported in bacteria [23] and proposed in a parallel study for mammalian cells [24]. In DN3 cells, the accumulation of RNA polymerase at the paused promoter could define a clear boundary, whereas the exact position of the elongating polymerase varies within each cell of a fixed DP population, generating a broader, more blurred boundary in the average map. While an attractive model, RNA polymerase II is absent from the inactive *Bcl6* gene in ESCs, where a relatively strong but somewhat broader border is nevertheless maintained at the promoter (**S7 Fig**), so other mechanisms must also contribute to architectural maintenance at this locus. Conserved CTCF binding at the promoter would be expected to be such a mechanism, although homozygous deletion of the major binding site had mild effects on locus architecture in ESCs (**S10 Fig**), in line with other studies suggesting that TADs are built up cooperatively from multiple elements [19,54,55]. We note also that the *Bcl6* gene locus contains multiple CTCF sites that are conserved in position in ESCs and thymocytes but do vary slightly in binding strength. Since clusters of CTCF are associated with "transition zones" between TADs [56], this may also influence architectural differences across cell type.

To directly test the effect of transcription on spatial domain border extension, we performed CRISPRa [42] to recruit the activator VP64-p65-Rta (VPR; [57]) specifically to the *Bcl6* promoter in ESCs. Despite a greater than 50-fold induction of the gene, which we estimate to elevate expression to comparable levels as DP thymocytes (**S2 Table**), there were negligible effects on the insulation score profile and, actually, a slightly reduced interaction of the *Bcl6* gene body with upstream regions (**Fig 3D–3F**). Transcription of the gene therefore does not appear to directly affect topological insulation at this gene. Alternatively, distal enhancer interactions may also play a role in defining spatial chromatin domains [19], and they have previously been reported to track from the promoter into the gene body, presumably accompanying the progress of the engaged RNA polymerase [58]. In line with this, virtual 4C from a putative interacting enhancer located approximately 230 kb upstream of *Bcl6* (denoted by an H3K27ac peak) shows thymocyte-specific interactions with the *Bcl6* promoter, with DP-specific persistence of interaction into the downstream coding region (**Fig 3G**). The enhancer is inactive and noninteracting in ESCs, so it does not affect topological organization regardless

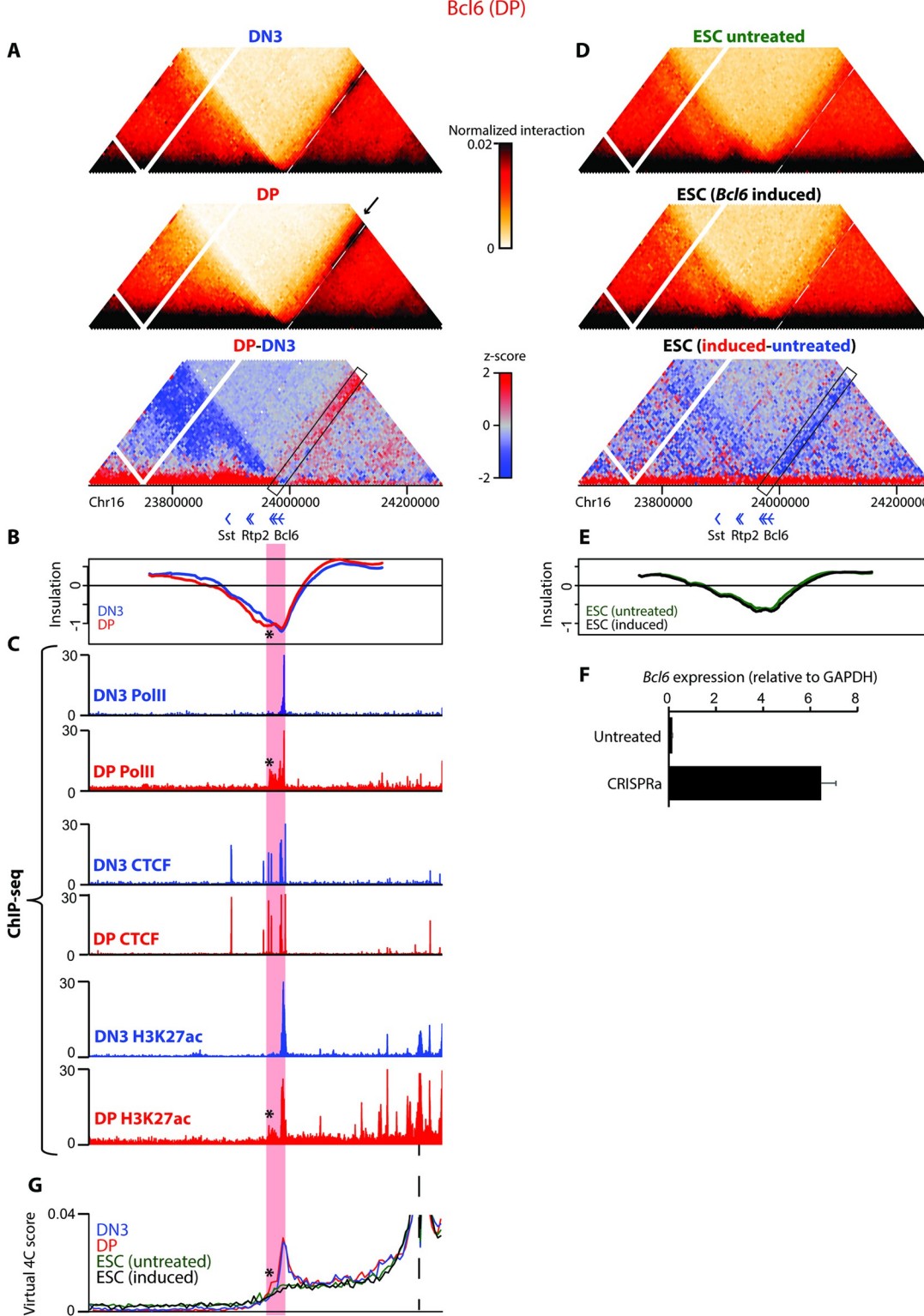

**Fig 3. Spatial domain border width at *Bcl6* gene tracks with RNA polymerase pause release.** (**A**) Pooled Capture Hi-C interaction maps for an approximately 600 kb region around the *Bcl6* gene are shown for DN3 and DP cells, above the differential heat map, where DP-enriched interactions are displayed in red and DN3-enriched interactions in blue. The white dotted lines represent the conserved strong border, with the arrows on the Hi-C maps and black rectangle on the differential map highlighting the broadening of the domain boundary in DP cells. Positions of genes are shown below the plots. (**B**)

Insulation scores for the same genomic region as in (**A**), computed at 2 kb resolution with a 100-kb (50 bin) window for DN3 (blue) and DP (red) cells. (**C**) ChIP-seq profiles (normalized as counts per million reads) for the same genomic region as in (**A**), for DN3 (blue) and DP (red) cells. Top to bottom: RNA polymerase II; CTCF; H3K27ac. Red stripe corresponds to the broadened domain boundary in DP cells, with corresponding extension of lower insulation score, RNA polymerase II binding, and H3K27ac loading into the gene body of *Bcl6* indicated by an asterisk. (**D**) Pooled Capture Hi-C interaction maps for the same genomic region as in (**A**) are shown for untreated ESCs and ESCs with ectopic activation of *Bcl6*, above the differential heat map. The white dotted lines representing the strong thymocyte domain border, and the arrows on the Hi-C maps and black rectangle on the differential map representing the broadening of the domain boundary in DP cells, have been included for reference. Positions of genes are shown below the plots. (**E**) Insulation scores for the same genomic region as in (**A**), computed at 2 kb resolution with a 100-kb (50 bin) window for untreated (green) and *Bcl6*-induced (black) ESCs. (**F**) Bar chart showing normalized *Bcl6* expression levels in untreated and CRISPRa-treated cells determined by qRT-PCR (2 biological replicates). (**G**) Virtual 4C plots derived from pooled Capture Hi-C datasets, using an upstream putative DP enhancer/H3K27ac peak as bait (position given by dashed line), for DN3 (blue), DP (red), untreated ESCs (green), and CRISPRa-treated ESCs (black). Red stripe corresponds to broadened domain boundary in DP cells, with corresponding increased interaction of the *Bcl6* gene body with the enhancer highlighted with an asterisk. Source data can be found in S1 Data. CRISPRa, CRISPR activation; DN3, double negative; DP, double positive; ESC, embryonic stem cell.

of *Bcl6* expression. It is worth noting that DN3 and DP cells have an identical positioning of the local insulation score minimum, at the *Bcl6* promoter, with an equivalent score in both thymocyte populations. Instead, the insulation score stays consistently lower in the *Bcl6* coding sequence in DP cells, which tracks well with both the DP-specific extension of RNA polymerase II and the extended region interacting with the upstream enhancer (* in **Fig 3B, 3 and 3G**). Such subtle insulation changes are missed from conventional TAD boundary calling approaches, so it is unclear whether the border broadening we observed at the *Bcl6* locus is a common occurrence in the genome. We did not observe this in any other region interrogated by the Capture Hi-C, and the resolution of our Hi-C data was insufficient to detect such subtle changes at the *Bcl6* gene or other loci in thymocytes.

## Transcription can directly remodel chromatin architecture

An additional type of topological change we observed when comparing thymocytes was the establishment of subdomains encompassing single genes, specifically in the cell type(s) where the gene is more highly expressed and contains most bound RNA polymerase II. In the regions targeted by Capture Hi-C, we observed a DP-specific spatial domain at the *Nfatc3* gene (**Figs 4A–4C** and **S4**) and a DN3-specific domain at the *Tmem131* gene (**S2** and **S11A–S11C Figs**), neither of which are accompanied by altered CTCF binding. To test whether transcription can be a direct driver of such chromatin architecture, we used CRISPRa to induce *Nfatc3* approximately 4-fold in ESCs where the subdomain is absent, bringing expression to approximately 80% of the level measured in DP cells (**S2 Table**) [59]. On induction, a topological domain encompassing the *Nfatc3* gene was identified in the Capture Hi-C map (**Fig 4D, 4E and 4I**), providing, to our knowledge, the first direct evidence that transcription can be instructive in spatial chromatin domain formation. Notably, the borders of the *Nfatc3* domain are also local minima of insulation score in conditions where the gene is silent, and either developmental (DN3-to-DP transition) or ectopic (CRISPRa in ESCs) induction of the gene does not greatly change insulation scores at these "proto-boundaries." Instead, the overall intradomain insulation score is increased, suggesting that transcriptional induction does not necessarily impede chromatin interactions between domains per se but rather reinforces intradomain contacts or compaction. As for *Bcl6*, these structural changes are largely overlooked by conventional insulation score analysis but can be detected by the alternative Hi-C analysis tool, CHESS (Comparison of Hi-C Experiments using Structural Similarity) [60]. This method successfully identified cell type–specific subdomains (alongside apparent false negatives) in the Capture Hi-C maps, but missed the widened border at *Bcl6* (**S12 Fig**), and was unable to find

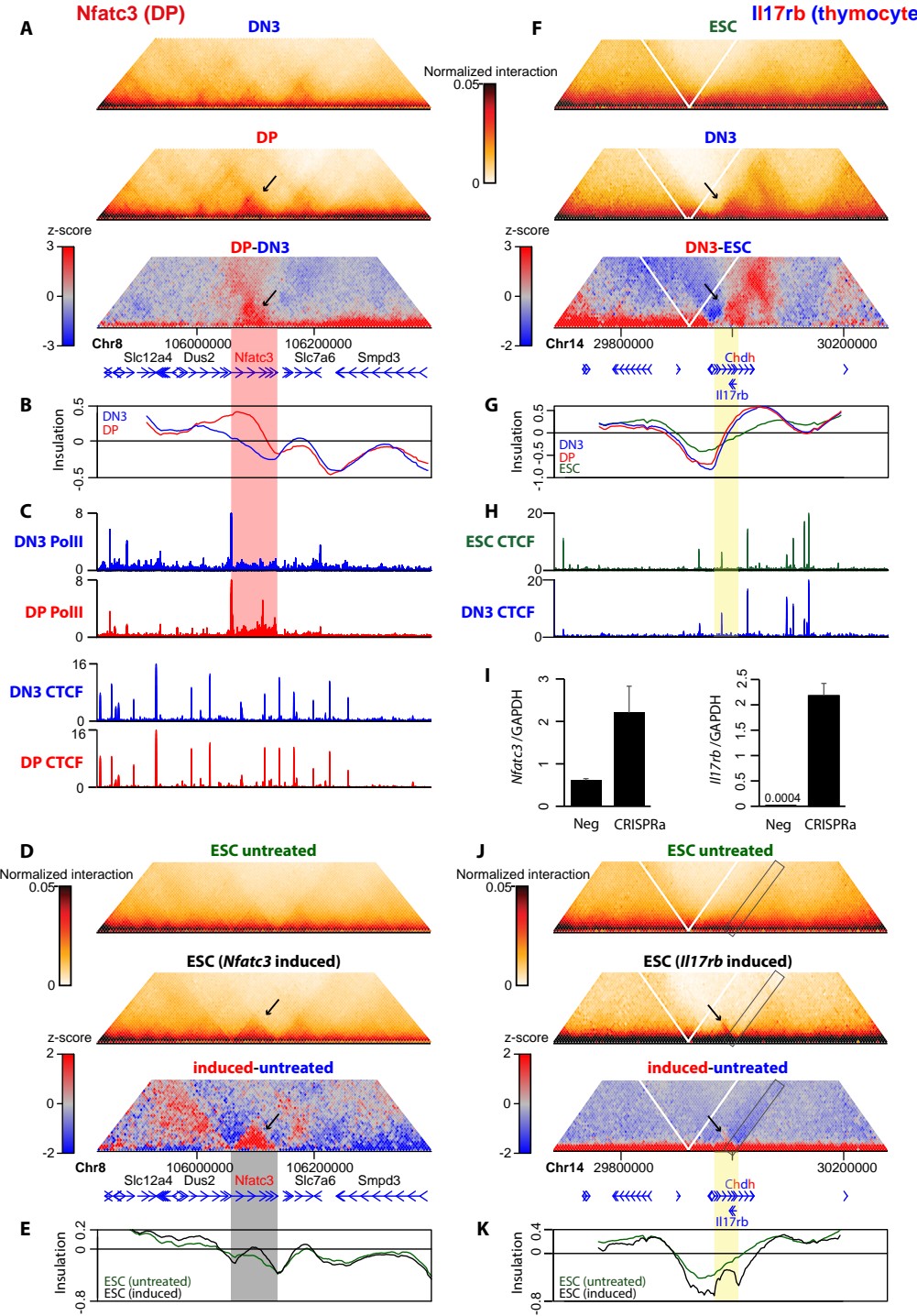

**Fig 4. Transcription can directly remodel chromatin architecture around activated genes.** (**A**) Pooled Capture Hi-C interaction maps for an approximately 600 kb region around the *Nfatc3* gene are shown for DN3 and DP cells, above the differential heat map, where DP-enriched interactions are displayed in red and DN3-enriched interactions in blue. The arrows indicate the DP-specific domain. Positions of genes are shown below the plots. (**B**) Insulation scores for the same genomic region as in (**A**), computed at 2 kb resolution with an 80-kb (40 bin) window for DN3 (blue) and DP (red) cells. (**C**) ChIP-seq profiles (normalized as counts per million reads) for the same genomic region as in (**A**), for DN3 (blue) and DP (red) cells. Top to bottom: RNA polymerase II, CTCF. Red stripe indicates DP-specific domain, with increased insulation score maximum, increased RNA polymerase II loading, and negligible changes in CTCF binding. (**D**) Pooled

Capture Hi-C interaction maps for the same genomic region as in (**A**) are shown for untreated ESCs and ESCs with ectopic activation of *Nfatc3*, above the differential heat map. Arrow indicates position of strengthened domain brought about by *Nfatc3* induction. (**E**) Insulation scores for the same genomic region as in (**A**), computed at 2 kb resolution with an 80-kb (40 bin) window for untreated (green) and *Nfatc3*-induced (black) ESCs. Dark stripe indicates position of reinforced domain with corresponding increase in insulation score maximum. (**F**) Pooled Capture Hi-C interaction maps for an approximately 600 kb region around the *Il17rb* gene are shown for ESCs and DN3 cells, above the differential heat map, where DN3-enriched interactions are displayed in red and ESC-enriched interactions in blue. The arrows indicate the thymocyte-specific domain, encompassing *Il17rb* and *Chdh*. Positions of genes are shown below the plots. (**G**) Insulation scores for the same genomic region as in (**F**), computed at 2 kb resolution with an 80-kb (40 bin) window for ESCs (green), DN3 (blue) and DP (red) cells. (**H**) CTCF ChIP-seq profiles (normalized as counts per million reads) for the same genomic region as in (**F**), for ESCs (green) and DN3 cells (blue). Yellow stripe indicates thymocyte-specific domain. (**I**) Bar charts showing normalized *Nfatc3* and *Il17rb* expression levels in untreated and CRISPRa-treated cells determined by qRT-PCR (2 biological replicates). (**J**) Pooled Capture Hi-C interaction maps for the same genomic region as in (**F**) are shown for untreated ESCs and ESCs with ectopic activation of *Il17rb*, above the differential heat map. Arrow indicates position of strengthened domain, and rectangle shows reduction in interdomain interactions brought about by *Il17rb* induction. (**K**) Insulation scores for the same genomic region as in (**F**), computed at 2 kb resolution with an 80-kb (40 bin) window for untreated (green) and *Il17rb*-induced (black) ESCs. Yellow stripe indicates position of new domain, containing new insulation score minima flanking an insulation score local maximum. Source data can be found in S1 Data. CRISPRa, CRISPR activation; DN3, double negative; DP, double positive; ESC, embryonic stem cell.

differential subdomains in the lower-resolution Hi-C maps. For *Tmem131*, the insulation scores of the boundaries did differ somewhat between thymocyte stages, but the increase in intradomain contacts was much more pronounced. Trials of multiple gRNAs failed to induce *Tmem131* in ESCs in CRISPRa trials, so we were unable to assess whether this domain was also directly remodeled by transcription.

Another gene within our own capture strategy, *Il17rb*, may have also been expected to comprise a DN3-specific domain: a cell type–specific transcribed region flanked by local insulation score minima (Figs **4G** and **S6**). Despite only negligible local interaction differences between DN3 and DP at this locus, closer inspection revealed that the reduced expression of *Il17rb* in DP cells is nonetheless approximately 40-fold higher than in ESCs (**S2 Table**). Differential Capture Hi-C heat maps revealed the presence of a *thymocyte*-specific contact domain, comprising *Il17rb* and the neighboring gene *Chdh*, again correlating with increased transcription of both genes (**Fig 4F–4H**). We used CRISPRa to induce *Il17rb* to comparable total expression levels as was obtained for *Nfatc3*, but this time representing an approximately 5,000-fold induction of the gene compared to almost completely silent wild-type ESCs, which we estimate to be approximately 40-fold greater than the level in wild-type DN3 thymocytes (and approximately 130-fold greater than wild-type DP thymocytes; **Fig 4I** and **S2 Table**). As may be expected, this overexpression had minimal effects on chromatin architecture around the neighboring *Chdh* gene but remodeled a new spatial domain at and just downstream of the induced *Il17rb* gene (**Fig 4J and 4K**), providing a second demonstration that transcription can directly remodel chromatin architecture. Compared to *Nfatc3*, which is an approximately 6-fold longer gene, the induced *Il17rb* domain was less apparent on the differential contact map, but the insulation score profile demonstrated a clear local maximum at the center of the novel domain, indicative of a similar intradomain compaction/interaction increase. Unlike on *Nfatc3* induction, where local insulation minima already demarcated the domain in wild-type ESCs, new domain boundaries were generated as shown by local insulation minima. The border downstream of *Il17rb* is at the same location as that found in thymocytes, and which is missing from wild-type ESCs. More strikingly, the border induced just upstream of the *Il17rb* promoter, also clearly visible as a "stripe" of reduced interdomain interactions (rectangle in **Fig 4J**), is unique to induced ESCs. The spatial domain extends further 3′ in thymocytes, presumably due to incorporation of the highly expressed *Chdh* gene, and is not apparent at all in wild-type ESCs.

Overall, these findings are reminiscent of previous observations of triptolide-sensitive single-gene domains in *Drosophila* [37] and of a correlation between RNA polymerase occupancy and intragene contacts in ESCs [35] in ultrahigh-resolution Hi-C or Micro-C maps, although neither study had assessed whether domains were directly formed as a consequence of transcription. Analysis of the thymocyte Hi-C datasets, restricted to sufficiently long genes to accommodate for the resolution limit, also showed an overall positive correlation between intragene contact strength and RNA polymerase occupancy, as well as other epigenetic marks of gene activity, such as H3K27ac and monomethylation of lysine-4 on histone H3 (H3K4me1) (**Fig 5A**). The RNA polymerase correlation was quantitatively very similar to what was observed in ESCs (Spearman correlation coefficient of 0.60 (35), compared to DN3 (0.62) and DP (0.61) in this study), and presence of the repressive histone modification, trimethylation of lysine-27 on histone H3 (H3K27me3), weakly anticorrelated with intragene contact strength. When clustering genes into 4 groups based on transcriptional output, metagene analysis further confirms the gradual increase of intragene compaction by augmenting gene expression (**Fig 5B**). Additionally, the most highly expressed thymocyte genes also demonstrated stronger interactions between transcription start sites and transcription termination sites in metagene analysis (rightmost column in **Fig 5B**), in line with previous suggestions of active gene looping events [39], although the resolution is insufficient to distinguish between point-to-point interactions between gene termini and general compaction of the entire gene body to a more homogeneous domain. These and the previously reported correlations apply to comparisons of gene sets within the same cell type. An advantage of our experimental setup is that we can also compare architectures for the same genes in different, but developmentally very close, cell types. When comparing intragene compaction *changes* between DN3 and DP cells, we also find a significant but weaker positive correlation with changes in RNA output, polymerase occupancy, and active histone modifications (Spearman correlation coefficient 0.21 to 0.30; *p*-value $< 1 \times 10^{-24}$) (**Fig 5C**). This suggests that, globally, gene up-regulation can remodel local spatial chromatin domains but is not necessarily a universal occurrence, with domains having different sensitivities. This is mirrored in our Capture Hi-C results when comparing spatial chromatin domains at *Il17rb* and *Tmem131*. Even though both genes form induced spatial domains, the apparent approximately 4-fold reduction in *Il17rb* expression on DN3-to-DP transition is insufficient to significantly disrupt the domain, whereas a smaller (approximately 2.5-fold) reduction in *Tmem131* expression causes major loss of architecture. Overall, our results strongly support that direct contact domain remodeling by transcription can occur, but with an apparent context dependence, which is hidden from previous global analyses.

## Discussion

In this work, we assessed whether chromatin architecture remodeling accompanied mouse thymocyte maturation, focusing at higher resolution on key, differentially expressed genes located close to strong TAD boundaries, where any such structural changes were most likely to occur. In line with lower-resolution studies [46], the majority of architectures appeared unchanged, despite significant transcriptional changes in hundreds of genes. Conventional measures of TAD architecture, such as the insulation score, were highly similar, both between thymocytes and unrelated cell types such as pluripotent cells, suggesting that most chromatin architecture at this scale is somehow "hard-wired" [49], fitting with evolutionary conservation of TAD positions at syntenic regions across species [61]. Physical models propose that TAD homeostasis may be largely explained by cohesin-mediated loop extrusion interfering with higher-order configurations such as coassociation of active compartments [62]. The latter are

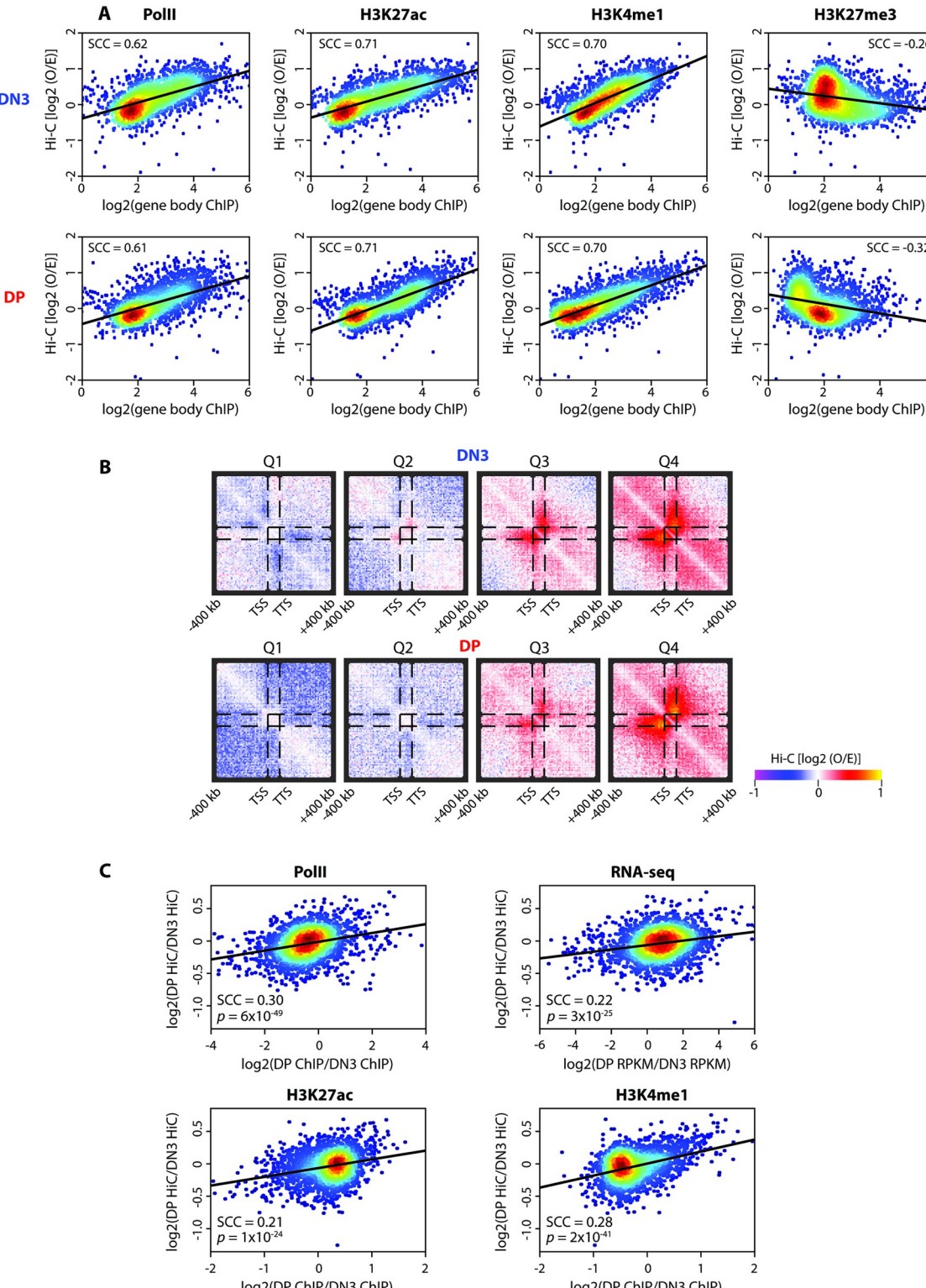

**Fig 5. Genome-wide intragene domain contacts correlate with hallmarks of transcription and upregulation.** (**A**) Scatter plots correlating intragene contacts as measured by Hi-C in DN3 (top) and DP (bottom) cells with median ChIP-seq signal on the gene for marks (from left to right: RNA polymerase II, H3K27ac, H3K4me1, H3K27me3). Spearman correlation coefficients are denoted on the plots. (**B**) Pileup Hi-C maps at 10 kb resolution showing the cumulative interaction between TSSs and TTSs in DN3 (top) and DP (bottom) cells with genes stratified according to expression levels, from the lowest quartile (Q1; left) to the highest quartile (Q4;

right). (**C**) Scatter plots for the ratios of intragene contact score relative to the ratio of median ChIP-seq gene signals between DN3 and DP cells. Spearman correlation coefficients and *p*-values are denoted on the graphs. Source data can be found in S1 Data. DN3, double negative; DP, double positive; TSS, transcription start site; TTS, transcription termination site.

dependent on underlying gene activity and epigenetic state so are developmentally plastic, as observed [21,49], whereas loop extrusion barriers could be genetically coded, such as by CTCF binding sites. Hence, in this model, TADs would not be expected to change significantly across cell type. However, CTCF occupancy is not identical in all cell types and in any case does not account for all spatial chromatin domains [7,19]. Recent studies have shown that non-encoded features as diverse as MCM complex binding (minichromosome maintenance; most known for replication origin licensing) and DNA double-strand breaks can also be barriers to loop extrusion [63,64]. Thus, small modifications to the basic "loop extrusion versus compartmentalization" model could just as easily explain tissue-specific spatial domains. An important challenge in the field will be to identify more comprehensively the features, both those genetically encoded/invariant and those more variable and linked to genomic functions, which can obstruct loop extrusion, and to assess their relative contribution to steady-state chromatin architecture.

Higher-resolution studies [21,35,37], including this one, do identify tissue-specific variations in spatial domains; they were presumably overlooked in previous works because they are too subtle and/or at too small a scale to be resolved. It is important to note that the most commonly utilized method to find TAD boundaries, identifying insulation score minima, does not detect the most important changes that we observed in this study. Expression-linked contact domains on genes and cell type–specific broadened borders often maintained the same local insulation minima in the cell types tested; instead, the insulation maxima were increased, or lower insulation scores extended beyond the minima, respectively. The alternative Hi-C analytical method CHESS [60] had some success in identifying these topological changes, but there is scope for development and application of new tools to find otherwise overlooked architectural changes. We provide a very convincing case that transcription can indeed play an instructive role in chromatin architecture, by recapitulating the subdomain around the *Nfatc3* gene and generating a novel gene-specific domain and border at the *Il17rb* gene, by their transcriptional induction in ESCs. While compelling, other genes in this and previous work provide counter-examples where transcriptional induction had negligible effects on domain organization [21]. Our study thus highlights 2 major questions for the field. Firstly, how exactly does transcription affect spatial chromatin domains? The main, nonmutually exclusive hypotheses already proposed are that bound RNA polymerase is a direct obstacle to loop extrusion [23] and that transcribed units form small A compartments disrupting the B compartments within which they reside [37]. Our results do not exclude either model and actually suggest that one may be prominent over the other in a context-dependent manner. Overexpression of *Il17rb* was able to generate a spatial domain border at the promoter, which lacks bound CTCF, whereas in the case of *Bcl6*, conventional CTCF-mediated architecture appears to play the major role, particularly in ESCs where there is no transcription. However, despite a similar CTCF binding profile at this locus for all 3 cell types studied, the boundary is much sharper in DN3 cells, corresponding exactly with the position of paused RNA polymerase. Release of this pause on full transcription in DP cells rewidens the boundary again, which could be explained if the polymerase acted as a mobile loop extrusion barrier. Similar gene body-broadened boundaries have been reported on metagene analysis in a parallel study [24], and we have recently proposed in fission yeast that paused or backtracking RNA polymerase may have different effects on condensin-mediated loop extrusion to elongating polymerase

[65]. Ectopic induction of the gene had no effect on local chromatin topology in ESCs, but the boundary was already rather wide, so the model is not necessarily disproved. However, the intriguing finding that the secondary boundary also tracks with apparent distal enhancer interactions raises the possibility of alternative gene looping mechanisms being involved. Further work on more loci at high resolution is required to tease out these different possibilities. On the other hand, transcriptional activation at the genes *Nfatc3* and *Tmem131* caused no apparent changes to the local insulation minima flanking these genes, which are much weaker candidate TAD boundaries than for *Bcl6*. Instead, the entire gene body appeared to form its own subdomain, in line with previous reports in yeast and ESCs [35,36]. In this case, there is no obvious means by which loop extrusion is being affected, since contacts between the gene and flanking regions do not appear to be changed. Rather, increased intragenic contacts suggest some looping or compaction of the gene unit. Similar transcription-coupled distortions of border activity were also reported in neuronal genes by an orthogonal study, albeit at relatively low resolution [66]. Since we were unable to fairly compare large-scale gene interactions with the rest of the chromosome in the Capture Hi-C experiments, it is unclear whether this local structural change is accompanied by a change in compartment. If it did, this would be consistent with the "compartmentalized domain" model proposed in *Drosophila* [37], since most fly genes would be too short to easily observe any accompanying intragenic compaction.

The second major question arising is why are transcription-mediated remodeling changes not more universally observed? Since we observed differences in relative sensitivities to domain remodeling (compare *Tmem131* and *Il17rb* domains during the DN3-to-DP transition), it is possible that they are masked at most loci by more dominant factors, such as CTCF-mediated TAD boundary definition. As more high-resolution micro- and Hi-C maps become available, especially in comparative studies of more developmentally related cell types, such as in this study, more cases of transcription-mediated changes may be discerned. In this case, we will be in a better position to assess which principles may influence spatial chromatin domain positioning and strength at each locus. In combination with more sophisticated single-molecule [14] and live imaging [16], going beyond basic principles to understanding the context-specific nature of chromatin architecture at specific loci is becoming a more achievable goal.

## Materials and methods

### Experimental design

Capture Hi-C and Hi-C experiments were performed in biological duplicates on purified mouse thymocytes or cultured ESCs before and after manipulations such as CRISPRa for specific genes or homozygous deletion of candidate CTCF binding sites. Detailed below, biological replicates were first assessed for reproducibility, before pooling the data for higher-resolution analysis. TAD analyses via computation of insulation scores was performed over multiple window sizes to ensure robustness of subsequent findings. 4C-seq experiments were performed as single experiments. qRT-PCR verification of CRISPRa gene induction was performed in 2 biological replicates, once for each induction experiment.

### Thymocyte isolation

Thymocytes were obtained from 4-week-old c57/Bl6 mouse (Charles River; equal numbers of males and females) thymus by FACS. Thymuses were dissected, gently homogenized, and filtered through a 70-μm strainer in cold PBE (0.5% BSA and 2 mM EDTA in 1X PBS). To purify DP cells, approximately 600 million cells were incubated with anti-mouse CD4-PE, anti-mouse CD8a-FITC, and anti-mouse CD3e-APC antibodies (eBioScience), before washing with cold PBE, adding DAPI, and purifying viable (DAPI$^-$) PE$^+$ FITC$^+$ APC$^+$ cells by FACS.

To purify DN3 cells, approximately 4 billion cells were first depleted of DP cells by incubating with rat anti-CD4 and rat anti-CD8 anti-sera (kind gift of Susan Chan), washing with cold PBE, and binding the DP cells to Dynabeads-sheep anti-rat IgG (Invitrogen). The supernatant was centrifuged to harvest the cells, which were then incubated with anti-mouse CD4-FITC, anti-mouse CD8a-FITC, anti-mouse CD3e-FITC, anti-mouse B220-FITC, anti-mouse CD11b-FITC, anti-mouse Ly-6G(Gr-1)-FITC, anti-mouse NK1.1-FITC, anti-mouse CD44-APC, and anti-mouse CD25-PE antibodies (eBioScience) and washed with PBE. DAPI was added and viable $PE^+APC^-FITC^-$ cells were purified by FACS.

## ESC culture

*Mus musculus* J1 ESCs (ATCC SCRC-1010) were grown on gamma-irradiated mouse embryonic fibroblast cells under standard conditions (4.5 g/L glucose-DMEN, 15% FCS, 0.1 mM nonessential amino acids, 0.1 mM beta-mercaptoethanol, 1 mM glutamine, 500 U/mL LIF, gentamicin) and then passaged onto feeder-free 0.2% gelatin-coated plates for at least 2 passages to remove feeder cells before treatments.

## CRISPRa

Twenty million cell batches of J1 ESCs were transfected each with 20 μg dCas9-VPR vector (Addgene #63798) and 20 μg plasmid constructed by the IGBMC molecular biology platform (available on request), containing optimized scaffold for 4 gRNAs alongside a puromycin resistance gene and mCherry reporter (gRNA sequences are given in **S9 Table**), with Lipofectamine 2000 according to the manufacturer's instructions. After 24 h, transfected cells were treated with 3 μg/mL puromycin and 1 mg/mL G418, and after another 24 h, mCherry-positive cells were sorted by FACS before immediate processing.

## Deletion of major *Bcl6* CTCF site

One million J1 ESCs were transfected with 1 μg of a plasmid constructed by the IGBMC molecular biology platform (available on request), containing optimized scaffold for 2 gRNAs alongside a Cas9-HF-EGFP construct and a puromycin resistance gene, with Lipofectamine 2000 according to the manufacturer's instructions. After 3 days, transfected cells were treated with 3 μg/mL puromycin for 24 h, then with 1 μg/mL puromycin for a further 24 h, before sorting individual GFP-positive cells into 96-well plates, pre-prepared with feeder cells, with FACS. Colonies were amplified and clones with homozygous deletions were identified by genomic DNA extraction and PCR screening with different primer pairs flanking and inside the expected deletion region. Sequences of the deletion were confirmed by TA-cloning and sequencing the PCR products. This ESC line is available on request.

## RNA isolation and gene expression analysis by RT-qPCR

Total RNA was extracted using the Nucleospin RNA kit (Macherey Nagel), and cDNA was prepared using SuperScript IV (Invitrogen) with random hexamers according to the manufacturer's instructions. Quantitative PCR was performed in a LightCycler 480 (Roche) using SYBR Green I master mix according to the manufacturer's instructions. Expression was normalized to *Gapdh* (primer sequences are given in **S10 Table**).

## Capture oligonucleotide design

The mouse genome (mm10) was digested in silico with *Dpn*II, and fragments were filtered to include those ≥140 bp and with GC content between 20% and 80%. To assess target

mappability, the genome was split in silico into 50 bp fragments and remapped using Bowtie [67] with a filter to only include uniquely mapping fragments. The *Dpn*II fragments were further filtered to only include those where ≥80% of the sequence is covered by this uniquely mapping reference point. Fragments within 600 kb of the target genes (**S1 Table**) were retained, and the 120 nt directly adjacent to the *Dpn*II sites were used for capture oligonucleotides (full capture design is given in **S11 Table**), synthesized as a SureSelect library (Agilent).

## Capture Hi-C

Capture Hi-C was essentially performed as in [68]. Five million aliquots of cells were fixed in 2% formaldehyde for 10 min, quenched with 125 mM cold glycine, then collected by centrifugation and washed with PBS. Cells were lysed in lysis buffer (10 mM Tris-HCl (pH 8), 100 mM NaCl, 0.2% NP-40, protease inhibitor cocktail) on ice for 30 min and nuclei collected by centrifugation and resuspended in *Dpn*II restriction buffer (NEBuffer 2 for *Hind*III Hi-C). ESCs were permeabilized by treatment with 0.4% SDS for 1 h at 37˚C, followed by 1.6% Triton X-100 for 1 h at 37˚C. Thymocytes were permeabilized by treatment with 0.8% SDS for 20 min at 65˚C, followed by 40 min at 37˚C, then with 3.3% Triton X-100 for 1 h at 37˚C. Nuclei aliquots were digested overnight at 37˚C with 1,500 U *Dpn*II (2,000 U *Hind*III for *Hind*III Hi-C), then cohesive ends were filled in with biotin tags by incubating for 90 min at 37˚C with 15 μM dATP, 15 μM dTTP, 15 μM dGTP, 15 μM biotin-14-dCTP, and 25 U DNA polymerase I Klenow fragment. Nuclei were collected by centrifugation, and in situ ligation was performed overnight at 16˚C in 500 μL volumes of 1X ligase buffer (NEB) with 20,000 U T4 DNA ligase, before de-crosslinking overnight at 65˚C with 0.75 mg/mL proteinase K. DNA was purified by RNase A treatment, phenol/chloroform extraction and isopropanol precipitation. Around 5 μg aliquots of DNA were sonicated to a fragment size of approximately 100 to 400 bp in a Covaris sonicator E220 and bound to Steptavidin MyOne T1 beads, following the manufacturer's instructions. The DNA was then end-repaired by incubating for 1 h at 20˚C with 0.8 mM dNTPs, 0.3 U/μL T4 DNA polymerase, 1 U/μL T4 polynucleotide kinase, and 0.1 U/μL DNA polymerase I Klenow fragment in ligase buffer, then A-tailed by incubating for 1 h at 37˚C with 200 μM dATP and 0.2 U/μL Klenow fragment (3′-5′ exo⁻) in NEBuffer 2 (NEB), then Illumina PE adapter was added by incubating overnight at 20˚C with 15 μM PE adapter and 2,000 U T4 DNA ligase in ligase buffer, washing the beads with appropriate buffers in between each treatment. Hi-C material was then amplified from the beads with 7 to 9 cycles of PCR using Herculase II Fusion DNA polymerase (Agilent) according to the manufacturer's instructions. Material at this stage was quantified on a BioAnalyzer (Agilent), and *Dpn*II and *Hind*III Hi-C material was sequenced (2 × 50 nt) on a HiSeq 4000 (Illumina) following the manufacturer's instructions). For Capture Hi-C, 750 ng *Dpn*II Hi-C material was concentrated in a SpeedVac vacuum concentrator and then target capture was performed using the SureSelect XT Target Enrichment system (Agilent) with the custom-designed library (**S11 Table**), following the manufacturer's instructions, and the library was then sequenced (2 × 50 nt) on a HiSeq 4000 (Illumina).

## 4C-seq

3C was performed on ESCs and thymocytes essentially as for the first steps of the Hi-C described above (up to DNA purification before sonication), except that the biotin fill-in step was omitted. The DNA was digested with 5 U/μg *Csp*6I at 37˚C overnight, then repurified by phenol/chloroform extraction and isopropanol precipitation. The DNA was then circularized by ligation with 200 U/μg T4 DNA ligase under dilute conditions (3 ng/μL DNA) and purified by phenol/chloroform extraction and isopropanol precipitation, before PCR amplification

with primers containing Illumina adapter sequence (sequences given in **S12 Table**). Excess primers were removed with SPRI beads, and the PCR products were assessed on a BioAnalyzer (Agilent), then sequenced ($1 \times 50$ nt) with a HiSeq4000 (Illumina).

## Hi-C analysis

For convenience, Hi-C data were processed using the FAN-C suite of tools [69], entailing read mapping to the mm10 genome assembly, mating pairs, attribution to restriction fragments with filtering of self-ligated fragments and PCR duplicates, matrix balancing, and computing insulation scores. For compatibility with Capture Hi-C plots, the data for the relevant regions were extracted using the *dump* tool and visualized as for the Capture Hi-C matrices.

## Capture Hi-C preprocessing

To ensure that the captured regions were treated independently of the noncaptured regions, the datasets were processed with custom scripts, as entailed in [70]. The initial steps are essentially the same as FAN-C, entailing read mapping with Bowtie [67] to the mm10 genome assembly, mating pairs, and attribution to restriction fragments with filtering of self-ligated fragments and PCR duplicates. Custom perl scripts filter the data to include only those where both ends of a paired read correspond to regions targeted by capture oligonucleotides and to bin these submatrices to fixed genomic bins. Each captured subregion was treated separately for matrix balancing with the Knight–Ruiz method [11]; R code kindly shared by Aleksandra Pękowska [71]. All plots in this study visualizing (Capture)-HiC matrices alongside ChIP-seq tracks and insulation scores were generated by custom R scripts.

## 4C-seq analysis

4C-seq data were mapped to the mm10, processed, normalized, and visualized using 4See [72], as in [19].

## (Capture) Hi-C analysis

The *cis*-decay plot for **Fig 1E** was obtained from 10 kb Hi-C data with the *expected* function of FAN-C. For the 5-kb Capture Hi-C data, the *cis*-decay plot was derived directly from the balanced submatrices at the captured regions, obtaining the distribution of normalized interaction scores at different genomic separations, and plotting the median values ± the interquartile range.

Insulation scores were essentially computed as in [50], determined by counting interactions along sliding windows off the Hi-C diagonal. Lower counts/weaker interactions imply greater topological insulation and, thus, a greater likelihood of the existence of a TAD border. This has the additional advantage of giving a TAD border "score" for all genomic intervals, rather than simply calling TADs using a more complex set of (often arbitrary) parameters. The only tunable parameter required in computing insulation is the sliding window size, altering the sensitivity to TADs of different sizes within the folded hierarchy [53]. Local insulation score minima are candidate TAD boundaries, and their *boundary scores* are computed as the difference in insulation score between the minima at the boundary and the adjacent bins (the *delta* in [50]), with higher scores indicating sharper, stronger TAD borders. For 10 kb Hi-C data, insulation score was computed with the *insulation* tool of FAN-C, using sliding windows of 70, 100, and 150 kb (7, 10, and 15 bins). Local minima were subsequently identified with the FAN-C tool *boundaries*, using a window of 3 bins for computing the boundary score and with no initial threshold filter for minimal boundary score. For Capture Hi-C, insulation score was

computed with custom R scripts on the balanced submatrices at 5 kb (for individual replicates) and 2 kb (for pooled data) resolutions, using sliding windows of 15, 25, 35, 50, and 75 kb (3, 5, 7, 10, and 15 bins), and 30, 40, 50, 80, and 100 kb (15, 20, 25, 40, and 50 bins), respectively. Windows of 3 bins were used for computing boundary scores for each of these insulation profiles. To identify the most robust TAD boundaries (e.g., as shown in **Fig 2D**), insulation score minima with a boundary score ≥0.1 in at least 2 different window sizes were maintained. Adjacent boundaries were merged and the overall boundary score for each boundary was calculated as the mean of the component scores, which were ≥0.1. These final called borders were intersected using the *GenomicRanges* R package to derive the Venn diagrams (e.g., **Fig 2G**). Distributions of insulation score were plotted as violin plots (e.g., **Fig 2A**) using the *ggplot2* package of R and were compared across cell type by two-tailed Kolgorov–Smirnov tests.

Biological replicates were compared by Spearman correlation coefficients of normalized interaction scores (for Capture Hi-C) and insulation scores (for Hi-C and Capture Hi-C). Principal component analysis was performed on the biological replicates of Capture Hi-C insulation scores using the *prcomp* function of R.

To derive the differential interaction maps (e.g., **Fig 3A**) the normalized scores of one matrix were subtracted from the other, and the difference was expressed as a z-score:

$$z = \frac{diff - mean(diff)}{standard\ dev(diff)}$$

Virtual 4C plots were derived as one row from the relevant Capture Hi-C normalized interaction score submatrix, applying smoothing with a running mean over 3 bins.

CHESS [60] was applied to each captured region independently after first converting the normalized submatrices to FAN-C objects. The *sim* function of CHESS was applied on sliding 40-bin (200 kb) windows, with steps of 1 bin (5 kb). The results of the submatrices were pooled, and regions with structural differences were filtered as having a signal-to-noise threshold >0.5 and a z-score for ssim, computed for the grouped dataset, <−1, as per recommendations in the original article [60].

## Metagene analysis

For all mouse genes of size ≥100 kb (≥10 bins in Hi-C), the intragenic interaction for each gene in each thymocyte population was computed as the median observed/expected score (defined as the Hi-C interaction normalized by the median Hi-C interaction level at the same genomic separation computed over the whole genome; see *cis*-decay curve in **Fig 1E**) for all intragenic pairwise bin-to-bin combinations. For the same gene set, the median ChIP signal within the gene body (taken over 10 kb bins) was also calculated and used to generate the scatter plots in **Fig 5A**. **Fig 5B** was made like in [35,37]. Genes ≥100 kb were grouped into 4 quartiles based on their expression levels, and, for each group, the median heatmap of observed/expected interaction scores is computed, by scaling genes to a pseudo-size of 10 bins to align all the transcription start and termination sites for the intragene part of heatmap and by considering extensions of 400 kb (40 bins) beyond the genes (upstream and downstream) for the rest. In **Fig 5C**, for each gene, the log2 ratio of the intragenic interaction score in DP and DN3 is plotted versus the corresponding log2 ratio of median ChIP-seq signal within the gene body.

## Epigenomic datasets

Publicly available thymocyte and ESC RNA-seq and ChIP-seq data were obtained from the Genome Expression Omnibus (see **S13 Table**). Fastq files were mapped to mm10 with

Bowtie2, and then normalized BigWig tracks (counts per million reads) were generated with DeepTools [73]. The tracks were plotted in R using the same y-axis scale when comparing across tissues.

## Supporting information

**S1 Fig. High-resolution interrogation of TADs in thymocytes with Capture Hi-C. (A)** ESC Hi-C maps (data taken from [21]) are shown at 5 kb resolution for approximately 600 kb regions surrounding the genes *Rag1* (left) and *Pla2g4a* (right), showing a TAD border near these genes. Below are shown the positions of genes and RNA-seq tracks (normalized counts per million reads; non-strand-specific) from DN3 (blue) and DP (red) cells, showing differential expression of the target genes between the thymocyte populations. **(B)** Capture Hi-C maps for both biological replicates in DN3 cells are shown at 5 kb resolution for an approximately 1.2 Mb region, including the genes *Nfatc3* and *Cdh1*, to show reproducibility. Positions of genes are shown. **(C)** Scatter plot for all normalized interaction scores from the 2 DN3 Capture Hi-C replicates. SCC is shown on the graph. **(D)** Pooled Capture Hi-C map for DP cells is shown at an approximately 600 kb region around the gene *Bcl6*. Arrow indicates a putative interaction between the *Bcl6* promoter and an upstream enhancer, identified by a peak of H3K27ac in the ChIP-seq track below. 4C-seq in DP cells using the *Bcl6* promoter as bait indicates sustained interactions over the broad upstream H3K27ac domain. Source data available in S1 Data. DN3, double negative; DP, double positive; ESC, embryonic stem cell; SCC, Spearman correlation coefficient; TAD, topologically associated domain.
(PDF)

**S2 Fig. Capture Hi-C results at *Zap70* locus.** Pooled Capture Hi-C maps are shown at 5 kb resolution for an approximately 600 kb region comprising the thymocyte-expressed *Zap70* and DN3-up-regulated *Tmem131* genes alongside epigenomic profiles for ESCs (green), DN3 (blue), and DP (red) cells. Top to bottom: ESC, DN3, and DP Capture Hi-C maps, positions of genes, DN3, and DP RNA-seq (normalized counts per million reads; non-strand-specific), ChIP-seq (normalized counts per million reads) for RNA polymerase II and CTCF. Arrows on map and blue stripe indicates strengthened spatial domain in DN3 cells around *Tmem131* gene, correlating with increased RNA polymerase binding and not associated with major changes in CTCF binding. Source data available in S1 Data. DN3, double negative; DP, double positive; ESC, embryonic stem cell.
(PDF)

**S3 Fig. Capture Hi-C results at *Pla2g4a* locus.** Pooled Capture Hi-C maps are shown at 5 kb resolution for an approximately 600 kb region comprising the DN3-up-regulated *Pla2g4a* gene alongside epigenomic profiles for ESCs (green), DN3 (blue), and DP (red) cells. Top to bottom: ESC, DN3, and DP Capture Hi-C maps, positions of genes, DN3, and DP RNA-seq (normalized counts per million reads; non-strand-specific), ChIP-seq (normalized counts per million reads) for H3K27ac and CTCF. Source data available in S1 Data. DN3, double negative; DP, double positive; ESC, embryonic stem cell.
(PDF)

**S4 Fig. Capture Hi-C results at *Nfatc3*/*Cdh1* locus.** Pooled Capture Hi-C maps are shown at 5 kb resolution for an approximately 1.2 Mb region comprising the DP-up-regulated *Nfatc3* and DN3-up-regulated *Cdh1* genes alongside epigenomic profiles for ESCs (green), DN3 (blue), and DP (red) cells. Top to bottom: ESC, DN3, and DP Capture Hi-C maps, positions of genes, DN3, and DP RNA-seq (normalized counts per million reads; non-strand-specific), ChIP-seq (normalized counts per million reads) for RNA polymerase II and CTCF. Arrow on

map and red stripe indicates strengthened spatial domain in DP cells around *Nfatc3* gene, correlating with increased RNA polymerase binding and not associated with major changes in CTCF binding. Source data available in S1 Data. DN3, double negative; DP, double positive; ESC, embryonic stem cell.
(PDF)

**S5 Fig. Capture Hi-C results at *Cd3* locus.** Pooled Capture Hi-C maps are shown at 5 kb resolution for an approximately 600 kb region comprising the thymocyte-expressed *Cd3* gene cluster alongside epigenomic profiles for ESCs (green), DN3 (blue), and DP (red) cells. Top to bottom: ESC, DN3, and DP Capture Hi-C maps, positions of genes, DN3, and DP RNA-seq (normalized counts per million reads; non-strand-specific), ChIP-seq (normalized counts per million reads) for H3K27ac and CTCF. Arrows on maps and yellow stripe indicates strengthened spatial domain in thymocytes around *Cd3*, compared to ESCs, correlating with increased CTCF binding at one border. Source data available in S1 Data. DN3, double negative; DP, double positive; ESC, embryonic stem cell.
(PDF)

**S6 Fig. Capture Hi-C results at *Il17rb* locus.** Pooled Capture Hi-C maps are shown at 5 kb resolution for an approximately 600 kb region comprising the DN3-up-regulated *Il17rb* gene and thymocyte-expressed *Chdh* gene alongside epigenomic profiles for ESCs (green), DN3 (blue), and DP (red) cells. Top to bottom: ESC, DN3, and DP Capture Hi-C maps, positions of genes, DN3, and DP RNA-seq (normalized counts per million reads; non-strand-specific), ChIP-seq (normalized counts per million reads) for H3K27ac and CTCF. Arrows on maps and yellow stripe indicates strengthened spatial domain in thymocytes around *Il17rb*, compared to ESCs, without apparent changes in CTCF binding at the new domain border. Source data available in S1 Data. DN3, double negative; DP, double positive; ESC, embryonic stem cell.
(PDF)

**S7 Fig. Capture Hi-C results at *Bcl6* locus.** Pooled Capture Hi-C maps are shown at 5 kb resolution for an approximately 600 kb region comprising the DP-up-regulated *Bcl6* gene alongside epigenomic profiles for ESCs (green), DN3 (blue), and DP (red) cells. Top to bottom: ESC, DN3, and DP Capture Hi-C maps, positions of genes, DN3, and DP RNA-seq (normalized counts per million reads; non-strand-specific), ChIP-seq (normalized counts per million reads) for RNA polymerase II and CTCF. Arrow on map and red stripe indicates DP-specific broadened TAD border, correlating with extension of RNA polymerase II into the *Bcl6* gene body and accompanied by only mild CTCF binding changes. Source data available in S1 Data. DN3, double negative; DP, double positive; ESC, embryonic stem cell; TAD, topologically associated domain.
(PDF)

**S8 Fig. TADs are largely conserved, with cell type–specific differences. (A)** Violin plots for distributions of insulation scores computed on pooled Hi-C datasets at 10 kb resolution, using a 150-kb (15 bin) window, on ESCs (green), DN3 (blue), and DP (red) cells. Whether analysis is restricted to the region targeted in the Capture Hi-C (top) or applied to the whole genome (bottom), ESCs have apparently more homogeneous insulation scores than thymocytes. **(B)** Scatter plots comparing insulation scores from pooled Hi-C data at 10 kb resolution. Top: comparing DN3 and DP cells using a 150-kb (15 bin) window; bottom: comparing ESCs and DP cells using a 70-kb (7 bin) window. SCC values are given on the graph. **(C)** Plots of first 2 principal components for insulation scores computed on biological replicates of Capture Hi-C datasets at 5 kb resolution, using a 25-kb (5 bin) window (left) or a 75-kb (15 bin) window

(right) for DN3 (blue), DP (red), untreated ESCs (green), and ESCs after either homozygous deletion of the major CTCF site at the *Bcl6* promoter (yellow) or CRISPRa ectopic induction of *Bcl6* (black), *Nfatc3* (gray), or *Il17rb* (cyan). (**D**) Pairwise Venn diagrams for overlap of called TAD boundaries from the pooled Capture Hi-C data of ESCs (green), DN3 (blue), and DP (red) cells. Source data available in S1 Data. CRISPRa, CRISPR activation; DN3, double negative; DP, double positive; ESC, embryonic stem cell; TAD, topologically associated domain.
(PDF)

**S9 Fig. Cell type–specific TADs uncovered by conventional Hi-C.** (**A**) Venn diagram for overlaps of called TAD boundaries from the pooled Hi-C data of ESCs (green; data taken from [21]), DN3 (blue), and DP (red) cells. (**B**) Pairwise Venn diagrams for the overlaps, as in (**A**). (**C**) Pooled Hi-C maps shown at 10 kb resolution for an approximately 2 Mb region around the thymocyte-expressed *Runx1* gene in ESCs (top; green), DN3 (middle; blue), and DP (bottom; red) cells, just above color-coded plots showing the positions and scores of called TAD boundaries, and CTCF ChIP-seq profiles (normalized by counts per million reads). Positions of genes are shown at the bottom of the plot. Red and blue bars denote the position of a thymocyte-specific domain formed around the *Runx1* gene, with a thymocyte-specific boundary corresponding to increased CTCF binding denoted by an asterisk. Triangles denote the orientation of the CTCF motifs at the border of this thymocyte-specific spatial chromatin domain. Source data available in S1 Data. DN3, double negative; DP, double positive; ESC, embryonic stem cell; TAD, topologically associated domain.
(PDF)

**S10 Fig. Mild effects on deletion of the major CTCF site at the *Bcl6* promoter in ESCs.** Capture Hi-C maps are shown at 5 kb resolution at an approximately 600 kb region around the *Bcl6* gene in wild-type ESCs (top) and those with homozygous deletion of the major CTCF site at the *Bcl6* promoter (middle), as well as the differential map (bottom) comparing the two. Below are shown the positions of genes, the position of the deletion (red), the ChIP-seq track for CTCF in ESCs, and the plot of computed insulation scores at 2 kb with a 100-kb (50 bin) window for wild-type (green) and ΔCTCF (black) ESCs. The orientations of the main CTCF motifs are indicated by arrowheads. Dashed lines show sites of altered insulation including a loss of insulation at the deleted site. Arrows on maps show where these insulation changes become apparent as stripes of relatively increased or decreased interactions. Source data available in S1 Data.
(PDF)

**S11 Fig. Transcriptional remodeling of chromatin architecture at the *Tmem131* gene.** (**A**) Pooled Capture Hi-C interaction maps for an approximately 600 kb region around the *Tmem131* gene are shown for DN3 and DP cells, above the differential heat map, where DP-enriched interactions are displayed in red and DN3-enriched interactions in blue. The arrows indicate the DN3-specific domain. Positions of genes are shown below the plots. (**B**) Insulation scores for the same genomic region as in (**A**), computed at 2 kb resolution with an 80-kb (40 bin) window for DN3 (blue) and DP (red) cells. (**C**) ChIP-seq profiles (normalized as counts per million reads) for the same genomic region as in (**A**), for DN3 (blue) and DP (red) cells. Top to bottom: RNA polymerase II, CTCF. Blue stripe indicates DN3-specific domain, with increased insulation score maximum, increased RNA polymerase II loading, and negligible changes in CTCF binding. Source data available in S1 Data.
(PDF)

**S12 Fig. CHESS can identify cell type–specific subdomains.** Differential Capture Hi-C maps between DP (enriched interactions in red) and DN3 (enriched interactions in blue) are shown for the captured regions: (**A**) *Zap70*; (**B**) *Pla2g4a*; (**C**) *Rag1*; (**D**) *Cd3e*; (**E**) *Il17rb*; (**F**) *Bcl6*; (**G**) *Nfatc3*/*Cdh1*. The black triangles denote the regions called as structurally different between the 2 cell types by CHESS, including the cell type–specific spatial domains around the *Tmem131* and *Nfatc3* genes (denoted by arrows). The number of sliding windows identified as structurally different is denoted in white text at each region. The widened domain border at *Bcl6*, denoted with an asterisk, was not identified by CHESS. Source data available in S1 Data. CHESS, Comparison of Hi-C Experiments using Structural Similarity; DN3, double negative; DP, double positive.
(PDF)

**S1 Table. Capture Hi-C strategy.** Overview of the genomic regions for which Capture Hi-C probes were designed, identified by the name of the gene over which they were centered. Red genes are more highly expressed in DP cells. Blue genes are more highly expressed in DN3 cells. Black genes are expressed equivalently in both thymocyte populations. Two captured regions were fused as one larger region (identified by *).
(DOCX)

**S2 Table. Cell type–specific expression of genes in this study.** Normalized RNA-seq expression values are given for the genes described in this study in ESCs, DN3, and DP cells, along with their ratios to highlight differential expression.
(DOCX)

**S3 Table. Overview of Hi-C and Capture Hi-C datasets presented in this study.**
(DOCX)

**S4 Table. Reproducibility of Capture Hi-C.** Spearman correlation coefficients between all biological replicates performed in this study.
(DOCX)

**S5 Table. Topological insulation similarities across cell type.** For pairwise combinations of the insulation scores, computed over different windows, their similarities are quantified with Spearman correlation coefficient, as well as their difference in distribution, as indicated by the *p*-values after two-tailed Kolgorov–Smirnov tests (KS p-val).
(DOCX)

**S6 Table. Topological insulation similarities genome-wide across cell type.** As **S4 Table**, but on the lower-resolution genome-wide Hi-C data. SCC values are given separately for the region targeted by Capture Hi-C and for the whole genome ("all").
(DOCX)

**S7 Table. Called TAD borders from Capture Hi-C experiments.** NA indicates where a TAD boundary was not called in that cell type. Provided as a separate Excel file.
(XLSX)

**S8 Table. Called TAD borders from Hi-C experiments.** NA indicates where a TAD boundary was not called in that cell type. Provided as a separate Excel file.
(XLSX)

**S9 Table. Sequences of gRNAs used in this study.**
(DOCX)

**S10 Table. Sequences of qRT-PCR primers used in this study.**
(DOCX)

**S11 Table. Oligonucleotides used in Capture Hi-C.** Provided as separate Excel file.
(XLSX)

**S12 Table. Sequences of 4C-seq primers used in this study.** Red sequence denotes Illumina sequencing adapters.
(DOCX)

**S13 Table. Previously published GEO datasets used in this study.**
(DOCX)

**S1 Data. Provided as Excel sheet, the first worksheet gives an overview of the source data type with accompanying GEO accession numbers for raw sequencing data.** Subsequent worksheets provide the source data for generating the relevant plots.
(XLSX)

## Acknowledgments

We thank Susan Chan for sharing reagents and Aleksandra Pękowska for sharing R code for Hi-C matrix balancing. Sequencing was performed by the IGBMC GenomEast platform, a member of the France Génomique consortium (ANR-10-INBS-0009). This study was made possible because of the IGBMC flow cytometry and molecular biology platforms. We thank PSMN (Pôle Scientifique de Modélisation Numérique) of the ENS de Lyon for computing resources.

## Author Contributions

**Conceptualization:** Sanjay Chahar, Yousra Ben Zouari, Daniel Jost, Tom Sexton.

**Formal analysis:** Yousra Ben Zouari, Hossein Salari, Tom Sexton.

**Funding acquisition:** Daniel Jost, Tom Sexton.

**Investigation:** Sanjay Chahar, Yousra Ben Zouari, Hossein Salari, Dominique Kobi, Manon Maroquenne, Cathie Erb, Anne M. Molitor, Audrey Mossler, Nezih Karasu.

**Methodology:** Sanjay Chahar, Dominique Kobi, Manon Maroquenne, Cathie Erb, Anne M. Molitor, Audrey Mossler, Nezih Karasu.

**Project administration:** Daniel Jost, Tom Sexton.

**Software:** Yousra Ben Zouari.

**Supervision:** Daniel Jost, Tom Sexton.

**Validation:** Dominique Kobi, Manon Maroquenne, Cathie Erb, Anne M. Molitor, Audrey Mossler, Nezih Karasu.

**Writing – original draft:** Sanjay Chahar, Yousra Ben Zouari, Hossein Salari, Dominique Kobi, Manon Maroquenne, Cathie Erb, Anne M. Molitor, Audrey Mossler, Nezih Karasu, Daniel Jost, Tom Sexton.

**Writing – review & editing:** Sanjay Chahar, Yousra Ben Zouari, Hossein Salari, Dominique Kobi, Manon Maroquenne, Cathie Erb, Anne M. Molitor, Audrey Mossler, Nezih Karasu, Daniel Jost, Tom Sexton.

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
