## [Editor Report · Decision Letter 0]

27 Apr 2023

Dear Tom, 

Thank you for submitting your manuscript entitled "Context-dependent transcriptional remodeling of spatial chromatin domains during differentiation" for consideration as a Research Article by PLOS Biology. Please accept my apologies for the delay in getting back to you as we consulted with an academic editor about your submission and obtained the previous reviews from eLife.

Your manuscript has now been evaluated by the PLOS Biology editorial staff, as well as by an academic editor with relevant expertise, and I am writing to let you know that we would like to invite a revision of your study.

However, before we can invite a revision, we need you to complete your submission by providing the metadata that is required for full assessment. To this end, please login to Editorial Manager where you will find the paper in the 'Submissions Needing Revisions' folder on your homepage. Please click 'Revise Submission' from the Action Links and complete all additional questions in the submission questionnaire.

Once your full submission is complete, your paper will undergo a series of checks in preparation for the decision. To provide the metadata for your submission, please Login to Editorial Manager (https://www.editorialmanager.com/pbiology) within two working days, i.e. by Apr 29 2023 11:59PM.

Kind regards,

Richard

Richard Hodge, PhD

Associate Editor, PLOS Biology

rhodge@plos.org

PLOS

---

## [Editor Report · Decision Letter 1]

2 May 2023

Dear Dr Sexton,

Thank you very much for submitting your manuscript "Context-dependent transcriptional remodeling of spatial chromatin domains during differentiation" for consideration as a Research Article at PLOS Biology. As you know, your manuscript and plan of revision have been evaluated by the PLOS Biology editors and by an Academic Editor with relevant expertise.

Based on your responses to the previous reviews at eLife, we would welcome re-submission of a revised version that takes into account the reviewers' comments according to your proposed revision plan. After discussions with the academic editor, we will not make the inclusion of the Micro-C data to provide a more genome-wide view essential for the revision, but we would encourage you to include this data if you are already in the process of conducting the experiments or if it would not take too long. 

We cannot make any decision about publication until we have seen the revised manuscript and your response to the reviewers' comments at eLife. We will then assess your revised manuscript with our Academic Editor aiming to avoid further rounds of peer-review, although might need to consult with the reviewers, depending on the nature of the revisions.

**IMPORTANT - SUBMITTING YOUR REVISION**

*Re-submission Checklist*

*Published Peer Review*

*PLOS Data Policy*

*Blot and Gel Data Policy*

Sincerely,

Richard

Richard Hodge, PhD

Associate Editor, PLOS Biology

rhodge@plos.org

PLOS

---

## [Editor Report · Decision Letter 2]

26 Oct 2023

Dear Dr Sexton,

Thank you for your patience while we considered your revised manuscript "Context-dependent transcriptional remodeling of spatial chromatin domains during differentiation" for publication as a Research Article at PLOS Biology. This revised version of your manuscript has been evaluated by the PLOS Biology editors and the Academic Editor.

Based on our Academic Editor's assessment of your revision, I am pleased to say that we are likely to accept this manuscript for publication, provided you satisfactorily address the following data and other policy-related requests that I have provided below (A-F):

(A) We would like to suggest the following modification to the title: 

““Transcription induces context-dependent remodeling of chromatin architecture during differentiation”

(B) You may be aware of the PLOS Data Policy, which requires that all data be made available without restriction: http://journals.plos.org/plosbiology/s/data-availability. For more information, please also see this editorial: http://dx.doi.org/10.1371/journal.pbio.1001797

-Supplementary files (e.g., excel). Please ensure that all data files are uploaded as 'Supporting Information' and are invariably referred to (in the manuscript, figure legends, and the Description field when uploading your files) using the following format verbatim: S1 Data, S2 Data, etc. Multiple panels of a single or even several figures can be included as multiple sheets in one excel file that is saved using exactly the following convention: S1_Data.xlsx (using an underscore).

-Deposition in a publicly available repository. Please also provide the accession code or a reviewer link so that we may view your data before publication. 

Figure 1A-E, 2A-F, 3A-G, 4A-K, S1A-D, S2, S3, S4, S5, S6, S7, S8A-C, S9C, S10, S11, S12A-G 

(C) Thank you for depositing the sequencing data in the GEO database (GSE218090). However, I note that the data is currently on hold and scheduled for release on Jan 1st 2024. We ask that you please make this data publicly available before publication.

(D) Thank you for depositing the new code that you have generated in Github (https://github.com/TomSexton00/Chahar_etal_analysis). Per journal policy, we ask that you please deposit the code into a repository that provides a DOI and long-term maintenance. Therefore, we would be grateful if you could please deposit the code that are in that Github account into Zenodo (https://zenodo.org/). Please ensure that the code is sufficiently well documented and reusable, and that your Data Statement in the Editorial Manager submission system accurately describes where your code can be found.

(E) Please also ensure that each of the relevant figure legends in your manuscript include information on *WHERE THE UNDERLYING DATA CAN BE FOUND*, and ensure your supplemental data file/s has a legend.

(F) Please ensure that your Data Statement in the submission system accurately describes where your data can be found and is in final format, as it will be published as written there. 

We expect to receive your revised manuscript within two weeks. 

*Published Peer Review History*

*Press*

Sincerely,

Richard

Richard Hodge, PhD

rhodge@plos.org

PLOS

---

## [Editor Report · Decision Letter 3]

9 Nov 2023

Dear Dr Sexton,

On behalf of my colleagues and the Academic Editor, Tom Misteli, I am pleased to say that we can accept your manuscript for publication, provided you address any remaining formatting and reporting issues. These will be detailed in an email you should receive within 2-3 business days from our colleagues in the journal operations team; no action is required from you until then. Please note that we will not be able to formally accept your manuscript and schedule it for publication until you have completed any requested changes.

PRESS

Best wishes,

Richard

Richard Hodge, PhD

rhodge@plos.org

PLOS
